# A monoclinic polymorph of sodium birnessite for ultrafast and ultrastable sodium ion storage

Hui Xia [1,2], Xiaohui Zhu [1,2], Jizi Liu[1,2], Qi Liu[3], Si Lan [1,2], Qinghua Zhang[4], Xinyu Liu[4], Joon Kyo Seo[5], Tingting Chen[1,2], Lin Gu [4] & Ying Shirley Meng [5]

Sodium transition metal oxides with layered structures are attractive cathode materials for sodium-ion batteries due to their large theoretical specific capacities. However, these layered oxides suffer from poor cyclability and low rate performance because of structural instability and sluggish electrode kinetics. In the present work, we show the sodiation reaction of $Mn_3O_4$ to yield crystal water free $NaMnO_{2-y-\delta}(OH)_{2y}$, a monoclinic polymorph of sodium birnessite bearing $Na/Mn(OH)_8$ hexahedra and $Na/MnO_6$ octahedra. With the new polymorph, $NaMnO_{2-y-\delta}(OH)_{2y}$ exhibits an enlarged interlayer distance of about 7 Å, which is in favor of fast sodium ion migration and good structural stability. In combination of the favorable nanosheet morphology, $NaMn_{2-y-\delta}(OH)_{2y}$ cathode delivers large specific capacity up to 211.9 mAh g$^{-1}$, excellent cycle performance (94.6% capacity retention after 1000 cycles), and outstanding rate capability (156.0 mAh g$^{-1}$ at 50 C). This study demonstrates an effective approach in tailoring the structural and electrochemical properties of birnessite towards superior cathode performance in sodium-ion batteries.

[1] School of Materials Science and Engineering, Nanjing University of Science and Technology, 210094 Nanjing, China. [2] Herbert Gleiter Institute of Nanoscience, Nanjing University of Science and Technology, 210094 Nanjing, China. [3] Department of Physics, City University of Hong Kong, Kowloon, Hong Kong, China. [4] Beijing National Laboratory for Condensed Matter Physics, Institute of Physics, Chinese Academy of Sciences, 100190 Beijing, China. [5] Department of NanoEngineering, University of California San Diego, La Jolla, CA 92093, United States. These authors contributed equally: Hui Xia, Xiaohui Zhu, Jizi Liu. Correspondence and requests for materials should be addressed to H.X. (email: xiahui@njust.edu.cn) or to L.G. (email: l.gu@iphy.ac.cn) or to Y.S.M. (email: shmeng@ucsd.edu)

odium-ion batteries (SIBs) with low cost and abundant raw material resources can be promising power sources to replace lithium-ion batteries (LIBs) for future large-scale energy storage applications[1–5]. Na-containing layered materials, with relatively simple synthetic process and high theoretical capacity[6], have attracted great interest as cathode materials for SIBs. Among them, layered $Na_xMnO_2$ with different polymorphs are particularly interesting due to abundant manganese source and their relatively high redox potentials as cathodes[2,5]. With a high Na content, $NaMnO_2$ has a large theoretical capacity of about 243 mAh g$^{-1}$ associated with the $Mn^{3+}/Mn^{4+}$ redox couple. Recently, several major layered polymorphs of $Na_xMnO_2$ including O3-type $NaMnO_2$ (e.g. monoclinic-$NaMnO_2$), P2-type $Na_xMnO_2$, and birnessite-$Na_xMnO_2 \cdot nH_2O$ were extensively studied as cathode materials for SIBs[2,7–10].

O3-type and P2-type $Na_xMnO_2$ tend to have phase transformations at low Na contents due to gliding of oxygen layers and Na$^+$ ion/vacancy-ordering transitions during sodiation/desodiation, resulting in limited cycle life and poor rate capability[5–7]. In addition, as $Mn^{3+}$-containing layered material, both O3-type and P2-type $Na_xMnO_2$ suffer from Jahn–Teller distortions, which cause structural deterioration and inferior performance. Cation substitution in O3 or P2 was found to be a general and effective strategy to suppress the phase transitions and improve the structural stability, resulting in significant enhancement in cycle performance. Although P2 phase possesses fast Na$^+$ diffusion due to its low-energy conduction pathways, the specific capacity of P2 phase is limited by its relatively lower Na content ($x < 2/3$) compared to O3 phase. For Na$^+$ ion diffusion in a layered structure, the diffusion barrier is highly dependent on the interlayer spacing as demonstrated in Li$^+$ or Na$^+$ diffusion in layered structure[11,12]. Birnessite-$Na_xMnO_2 \cdot nH_2O$, with a much large interlayer spacing (~7 Å) compared to P2 (~5 Å) and O3 (~5 Å), is superior layered structure for fast Na storage. Compared with the O3-type and P2-type $Na_xMnO_2$, birnessite-$Na_xMnO_2 \cdot nH_2O$ is very stable with no phase transition in the charge/discharge processes because the crystal water can function as interlayer pillars to stabilize the layered structure, resulting in suppressed Jahn–Teller distortions[10]. Nevertheless, the interlayer crystal water also has an adverse effect on the electrochemical performance for the birnessite-$Na_xMnO_2 \cdot nH_2O$. Although the expanded interlayer space could facilitate fast Na$^+$ diffusion, the existence of crystal water at interlayer could hinder Na$^+$ diffusion and limit Na accommodation. In addition, the crystal water could be extracted during charge process and has side reactions with the electrolyte, thus deteriorating the electrochemical performance. By far, the Na content ($x$) in birnessite is still limited to about 0.71 for $Na_{0.71}MnO_2 \cdot 0.25H_2O$[10], which limits the maximum capacity that can be obtained from this material. It could be an ideal layered structure that possesses large interlayer spacing as birnessite but does not contain crystal water, which has not yet been successfully developed for $NaMnO_2$.

Herein, we report a monoclinic polymorph of Na-birnessite, $NaMnO_{2-y-\delta}(OH)_{2y}$ with large interlayer spacing of about 7 Å. The monoclinic $NaMnO_{2-y-\delta}(OH)_{2y}$ was obtained via annealing a high Na content birnessite-$NaMnO_{2-y}(OH)_{2y} \cdot 0.10H_2O$ in Ar atmosphere to remove the crystal water. The high Na content birnessite-$NaMnO_{2-y}(OH)_{2y} \cdot 0.10H_2O$ was reported for the first time and was obtained via a two-step "hydrothermal sodiation" on $Mn_3O_4$ nanowall arrays. The monoclinic $NaMnO_{2-y-\delta}(OH)_{2y}$ retains the large interlayer spacing due to the coexistence of H′3 stacking with eight-coordinate sites and O′3 stacking with six-coordinate sites in the layered structure. Oxygen vacancies were also introduced into the lattice during annealing in Ar atmosphere, which greatly increased the electrical conductivity of the layered material. It has been demonstrated that the new

monoclinic polymorph possesses ultrastable layered structure with superfast Na$^+$ migration. In specific, the monoclinic $NaMnO_{2-y-\delta}(OH)_{2y}$ can deliver a large reversible capacity of 211.9 mAh g$^{-1}$ at 0.2 C and retain 156.0 mAh g$^{-1}$ at 50 C, revealing excellent rate performance. A very high capacity retention of 94.6% for 1000 cycles has been achieved for the O′3/H′3 $NaMnO_{2-y-\delta}(OH)_{2y}$, demonstrating outstanding structural stability and ultralong cycling life. This work provides important insights in structural regulation to develop ideal layered material as promising cathode for SIBs.

## Results

**Materials synthesis and characterization.** The self-supported porous birnessite-$Na_xMnO_{2-y}(OH)_{2y}$ and monoclinic-$NaMnO_{2-y-\delta}(OH)_{2y}$ nanowall arrays were grown on carbon cloth by "hydrothermal sodiation" on the porous $Mn_3O_4$ nanowall arrays and the synthesis procedure is illustrated in Fig. 1a. The porous $Mn_3O_4$ nanowall arrays were deposited on the carbon cloth by cathodic deposition according to our previous work[13]. As shown in Fig. 1a, the hydrothermal treatment can enable the chemical sodiation to the preformed $Mn_3O_4$ nanowall arrays and convert them into birnessite-$Na_xMnO_{2-y}(OH)_{2y} \cdot nH_2O$. By controlling the hydrothermal temperature, S1 ($Na_{0.46}MnO_{2-y}(OH)_{2y} \cdot 0.61H_2O$), S2 ($Na_{0.71}MnO_{2-y}(OH)_{2y} \cdot 0.32H_2O$), and S3 ($NaMnO_{2-y}(OH)_{2y} \cdot 0.10H_2O$) samples with various Na and crystal water contents can be obtained. Importantly, the birnessite-$Na_xMnO_{2-y}(OH)_{2y} \cdot 0.10H_2O$ with a high Na content of $x = 1$ was first time prepared via the two-step "hydrothermal sodiation". The monoclinic-$NaMnO_{2-y-\delta}(OH)_{2y}$ (S4) can be easily obtained by annealing the birnessite-$NaMnO_{2-y}(OH)_{2y} \cdot 0.10H_2O$ in Ar atmosphere to remove the crystal water and induce oxygen vacancies. The Na/Mn atomic ratios of the S1–S4 products were determined by inductively coupled plasma (ICP) measurements (Supplementary Table 1).

Figure 1b shows the X-ray diffraction (XRD) patterns of the $Mn_3O_4$ nanowall arrays and the obtained products of S1–S4 after "hydrothermal sodiation". For the XRD pattern of the $Mn_3O_4$ nanowall arrays, except for two peaks at 26.5° and 44.2° assigned to carbon fabric, all other diffraction peaks can be assigned to spinel $Mn_3O_4$ phase (JCPDS Card no. 24-0734). All S1–S4 products exhibit similar XRD patterns, resembling a crystal structure similar to birnessite-$Na_{0.55}Mn_2O_4 \cdot 1.5H_2O$ (JCPDS Card no. 43-1456) with the space group of $C2/m$. Obviously, the (001) diffraction peak of the S4 sample shifts slightly to a higher angle, corresponding to a slight decrease of interlayer distance to ~6.9 Å compared to ~7.2 Å of S1–S3 samples. The shrinkage of interlayer distance of S4 can be ascribed to the removal of interlayer crystal water. The crystal water content in S1–S4 can be determined by thermogravimetric analysis (TGA, Supplementary Fig. 1 and Table 1). Crystal water can be completely removed above 250 °C and the crystal water content decreases as the Na content increases in the S1–S3 samples. The interesting phenomenon is that the interlayer distance of S4 is much larger than those (~5.4 Å) of O3-$Na_xMnO_2$ and P2- $Na_xMnO_2$ without crystal water. If not using the "hydrothermal sodiation" method developed in present study, the heat treatment on birnessite-$Na_xMnO_2 \cdot nH_2O$ prepared by other solution methods using $Mn^{2+}$ or $Mn^{7+}$ precursors[10] will result in a significant shrinkage of the interlayer distance from 7.2 to 5.4 Å (Supplementary Fig. 2). Raman measurements were further carried out to reveal complementary structural information to the spinel $Mn_3O_4$ and S1–S4 samples in Fig. 1c. The Raman spectrum of the $Mn_3O_4$ sample shows a sharp peak at 668 cm$^{-1}$, corresponding to the Mn–O breathing vibration of $Mn^{2+}$ ions in tetrahedral coordination[14]. Other small peaks at 290, 322, and 372 cm$^{-1}$ can be attributed to the $T2g(1)$, $Eg$, and $T2g(2)$ modes of $Mn_3O_4$,

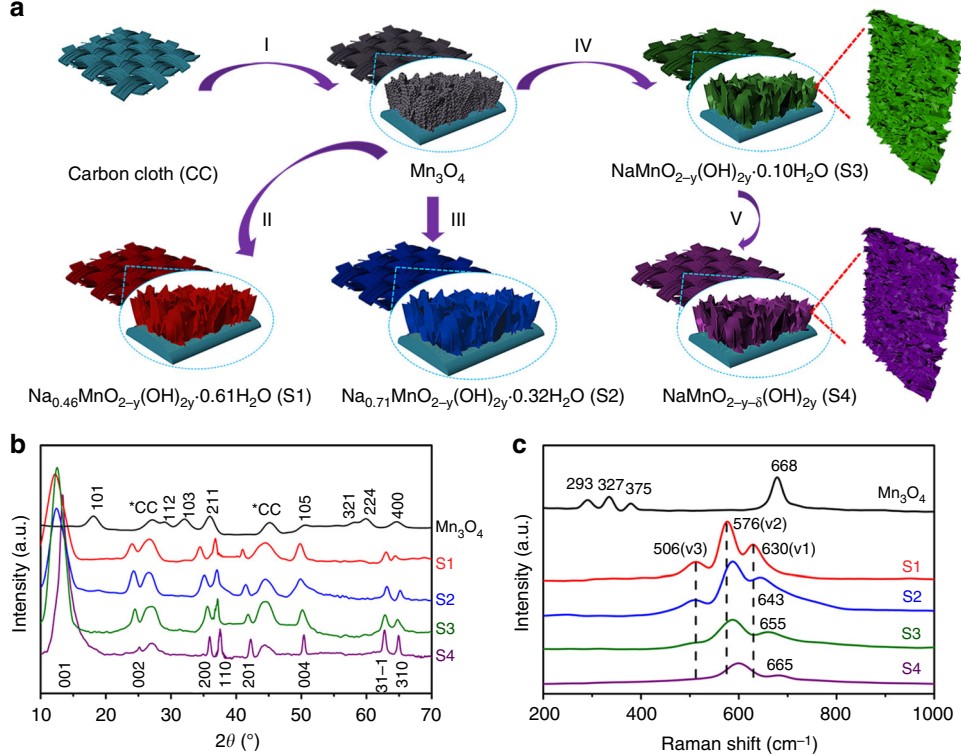

**Fig. 1** Synthesis and Structures of $Mn_3O_4$ and S1–S4 samples. **a** Schematic illustration of fabrication procedure of the porous S1–S4 nanowall arrays on carbon cloth (I, cathodic deposition of $Mn_3O_4$; II–IV, hydrothermal treatments at different conditions; V, annealing in Ar atmosphere). **b** XRD patterns of the $Mn_3O_4$ and S1–S4 samples. **c** Raman spectra of the $Mn_3O_4$ and S1–S4 samples

**Table 1 Comparison of electrochemical performances for different layered $Na_xMO_2$ electrodes (M, transition metal)**

| Cathode electrodes (loading mass/mg cm$^{-2}$) | Potential range (V) | Rate performance | Capacity retention | Reference |
|---|---|---|---|---|
| Monoclinic-NaMnO$_2$ (unknown) | 2.0–3.8 | 185 mAh g$^{-1}$ at 0.1 C | 132 mAh g$^{-1}$ after 20 cycles at 0.1 C | ref. [7] |
| β-NaMnO$_2$ (4–5) | 2.0–4.2 | 90 mAh g$^{-1}$ at 10 C | 65 mAh g$^{-1}$ after 100 cycles at 10 C | ref. [8] |
| Bir-Na$_{0.71}$MnO$_2$·0.25H$_2$O (1.1) | 1.85–3.65 | 89 mAh g$^{-1}$ at 8/3 C | 105 mAh g$^{-1}$ after 100 cycles at 0.4 C | ref. [10] |
| NaMn$_3$O$_5$ (~1.95) | 1.5–4.7 | 115 mAh g$^{-1}$ at 5 C | 153 mAh g$^{-1}$ after 20 cycles at 0.1 C | ref. [34] |
| P2/O′3-NaMnTi$_{0.1}$Ni$_{0.1}$O$_2$ (~1.3) | 1.5–4.2 | 103 mAh g$^{-1}$ at 10 C | 97 mAh g$^{-1}$ after 500 cycles at 5 C | ref. [35] |
| O3-Na$_{0.9}$[Cu$_{0.22}$Fe$_{0.30}$Mn$_{0.48}$]O$_2$ (3–5) | 2.5–4.05 | 59 mAh g$^{-1}$ at 5 C | 95 mAh g$^{-1}$ after 100 cycles at 0.1 C | ref. [36] |
| O3-NaNi$_{0.5}$Mn$_{0.2}$Ti$_{0.3}$O$_2$ (3–5) | 2.0–4.0 | 135 mAh g$^{-1}$ at 1 C | 115 mAh g$^{-1}$ after 200 cycles at 1 C | ref. [37] |
| P2-Na$_{0.6}$[Cr$_{0.6}$Ti$_{0.4}$]O$_2$ (unknown) | 2.5–3.85 | 61 mAh g$^{-1}$ at 2 C | 60 mAh g$^{-1}$ after 200 cycles at 1 C | ref. [38] |
| O3-NaNi$_{0.45}$Cu$_{0.05}$Mn$_{0.4}$Ti$_{0.1}$O$_2$ (3) | 2.0–4.0 | 81 mAh g$^{-1}$ at 10 C | 84 mAh g$^{-1}$ after 500 cycles at 1 C | ref. [39] |
| C-NaCrO$_2$ (3.5) | 2.0–3.6 | 106 mAh g$^{-1}$ at 50 C | 109 mAh g$^{-1}$ after 300 cycles at 0.5 C | ref. [40] |
| P2-Na$_{0.7}$CoO$_2$ (~1.4) | 2.0–3.8 | 64 mAh g$^{-1}$ at 16 C | 84 mAh g$^{-1}$ after 300 cycles at 4 C | ref. [41] |
| NaMnO$_{2-y-\delta}$(OH)$_{2y}$ (2–3) | 2.0–4.0 | 156 mAh g$^{-1}$ at 50 C | 174 mAh g$^{-1}$ after 1000 cycles at 10 C | This work |

respectively. The Raman spectra of the S1–S4 samples show three major peaks between 500 and 700 cm$^{-1}$, corresponding to the $v3$ (Mn–O) stretching vibration, $v2$(Mn–O) stretching vibration, and $v1$(Mn–O) symmetric stretching vibration in the basal plane of edge-shared MnO$_6$ octahedra, respectively[15]. With the increasing of Na content in the birnessite-Na$_x$MnO$_{2-y}$(OH)$_{2y}$, $v1$ and $v2$ vibrations of MnO$_6$ groups shift to higher energies[16], indicating the decrease of the average Mn oxidation state. For the S4 sample, $v1$ and $v2$ vibrations locate at higher energies compared to those of the S3 sample, implying even lower Mn oxidation state of the S4 sample. Due to the same Na content in both S3 and S4 samples, the lower average Mn oxidation state suggest the existence of O vacancies in the S4 sample. As shown in Supplementary Fig. 3a (Standard X-ray absorption near edge structure (XANES) spectra of Mn L$_{II,III}$-edges for MnO and Mn$_2$O$_3$ are also included as reference), the Mn L$_{II,III}$-edge shifts to

lower energy for S4 as compared to that of S3, demonstrating decrease of average Mn valence in S4. Differences in the O K-edge fine structure can be observed between the S3 and S4 spectra as revealed in Supplementary Fig. 3b. The O K-edge pre-peak is significantly reduced in the spectrum for S4 as compared to that of S3. The decrease in intensity of the pre-peak can be ascribed to the reduction of neighboring Mn and formation of Oxygen vacancies[17]. Mn reduction comes along with oxygen vacancy formation because of charge compensation. X-ray photoelectron spectroscopy (XPS) results further confirm the Mn oxidation state change in Mn$_3$O$_4$ and S1–S4 samples (Supplementary Fig. 4). The energy difference between the two peaks (or peak separation) in the Mn $3s$ core-level XPS spectra is correlated to the average Mn valence. From S1 to S4, the peak separation keeps increasing, indicating the average Mn valence is decreasing. Different peaks can be detected from the Mn $2p$ core-level XPS spectra of S1–S4,

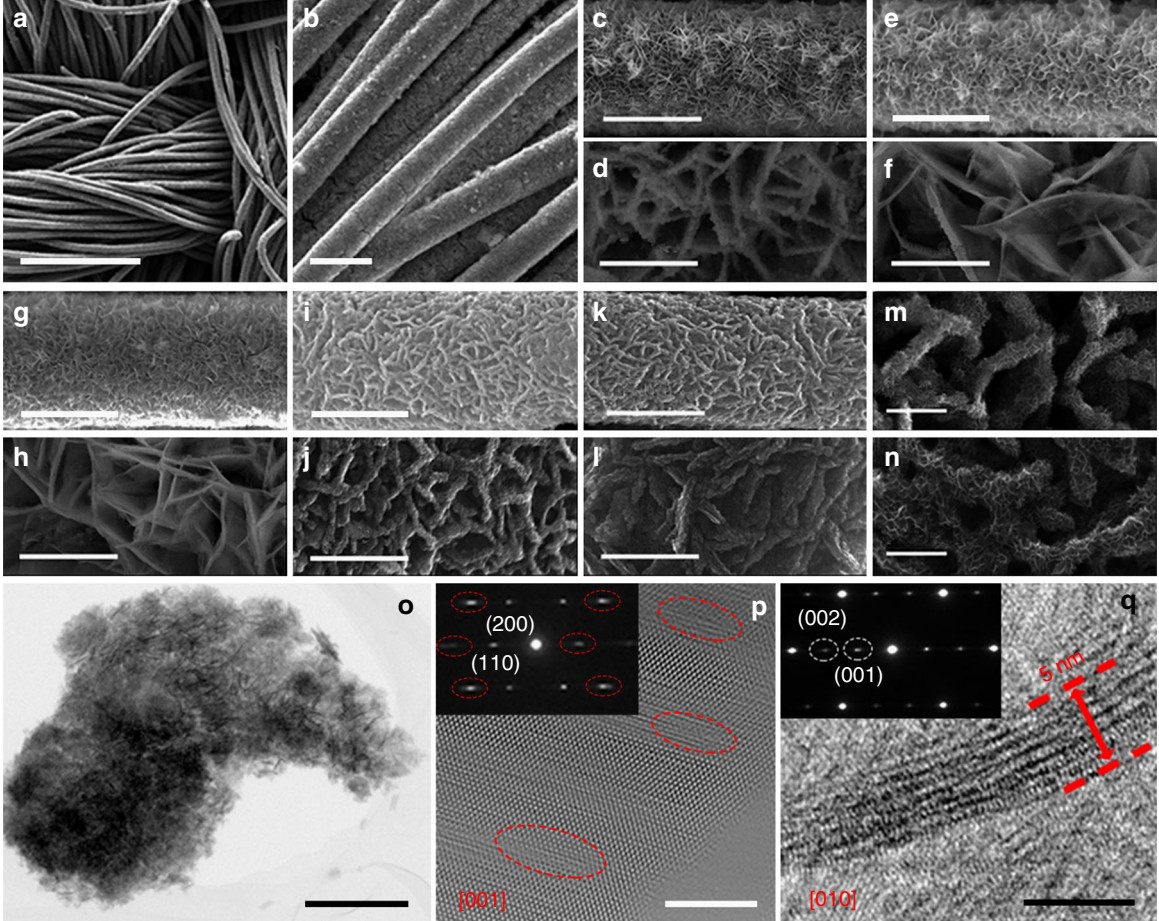

**Fig. 2** FESEM and TEM images. **a**, **b**, **k**, **l**, **n** FESEM images of the S4 sample. **c**, **d** FESEM images of the $Mn_3O_4$ nanowall arrays. **e**, **f** FESEM image of the S1 sample. **g**, **h** FESEM image of the S2 sample. **i**, **j**, **m** FESEM images of the S3 sample. **o** TEM image for the S4 sample. **p** HRTEM image for the S4 sample along [001] zone axis. Inset in **p** is the SAED pattern along [001] zone axis. **q** HRTEM image for S4 sample along [010] zone axis. Inset in **q** is the SAED pattern along [010] zone axis. **a** Scale bar, 150 μm. **b** Scale bar, 20 μm. **c**, **e**, **g**, **i**, **k** Scale bar, 10 μm. **d**, **f**, **h**, **j**, **l** Scale bar, 2 μm. **m**, **n** Scale bar, 500 nm. **o** Scale bar, 300 nm. **p**, **q** Scale bar, 5 nm

where peaks located at 642.3, 641.1 and 640.2 eV correspond to $Mn^{4+}$, $Mn^{3+}$, $Mn^{2+}$, respectively. The emergence of $Mn^{2+}$ in S4 sample further confirms the existence of oxygen vacancies after heat treatment in the Ar atmosphere. As shown in the O 1*s* core-level XPS spectra, a component corresponding to $Mn^{3+}$–O–H species can be found for all S1–S4 samples. No crystal water exists in the S4 sample as confirmed by the TGA result, suggesting the existence of –OH bonds in S4 even without crystal water in the lattice.

The surface morphologies of different samples are characterized using field emission scanning electron microscope (FESEM). Fig. 2a and b show the FESEM images of the S4 sample at low magnifications, revealing uniform growth of the nanowall arrays on the surface of carbon fabric in large scale. Fig. 2c and k show the FESEM images of the $Mn_3O_4$ and S4 nanowall arrays on a single carbon fiber, revealing that the nearly vertically aligned nanowalls are cross-linked, creating a highly porous shell on carbon fiber. Fig. 2d, f, h, j, and l illustrate the high-magnification FESEM images of the $Mn_3O_4$, S1, S2, S3, and S4 nanowalls arrays grown on a single carbon fiber, respectively. The $Mn_3O_4$ nanowalls are composed of numerous nanoparticles, thus exhibiting rough surface. The S1 and S2 samples retain the similar nanowall morphology, but the nanowalls are very smooth with transparent texture. The S3 and S4 samples exhibit obvious increase in the nanowall thickness with very rough surface. The enlarged FESEM images in Fig. 3m and n show that each

expanded nanowall is composed of interconnected nanosheets of ~5 nm in thickness. Fig. 2o displays the transmission electron microscopy (TEM) image of one $NaMnO_{2-y-\delta}(OH)_{2y}$ nanowall with low magnification, presenting a mesoporous structure. Powdery samples can be obtained from the carbon fabric support by ultrasonication. The nitrogen adsorption–desorption isotherms of the S4 powdery sample was measured and shown in Supplementary Fig. 5 with inserted pore size distribution curve, indicating typical mesoporous structure. The pore size distribution curve presents a relatively narrow pore size distribution around 9 nm. According to the BET computational method, the specific surface area of the S4 sample is about 105.3 $m^2 g^{-1}$. The scanning transmission electron microscopy (STEM) and corresponding energy dispersive X-ray spectroscopy (EDS) mapping images for the as-prepared S4 sample are shown in Supplementary Fig. 6, reveling uniform distributions of Na, Mn, and O elements in the sample. The high-resolution transmission electron microscopy (HRTEM) images of one $NaMnO_{2-y-\delta}(OH)_{2y}$ nanosheet recorded along the [001] and [010] zone axes are shown in Fig. 2p, q, respectively. The inserted selected area electron diffraction (SAED) patterns in Fig. 2p and q indicate single crystalline feature of the nanosheet. The elongated diffraction dots (marked by red circles) in the SAED pattern and different contrasts (marked by red circles) in the HRTEM image in Fig. 2p indicate existence of stacking faults along [001] orientation. Fig. 2q displays the cross-section HRTEM image

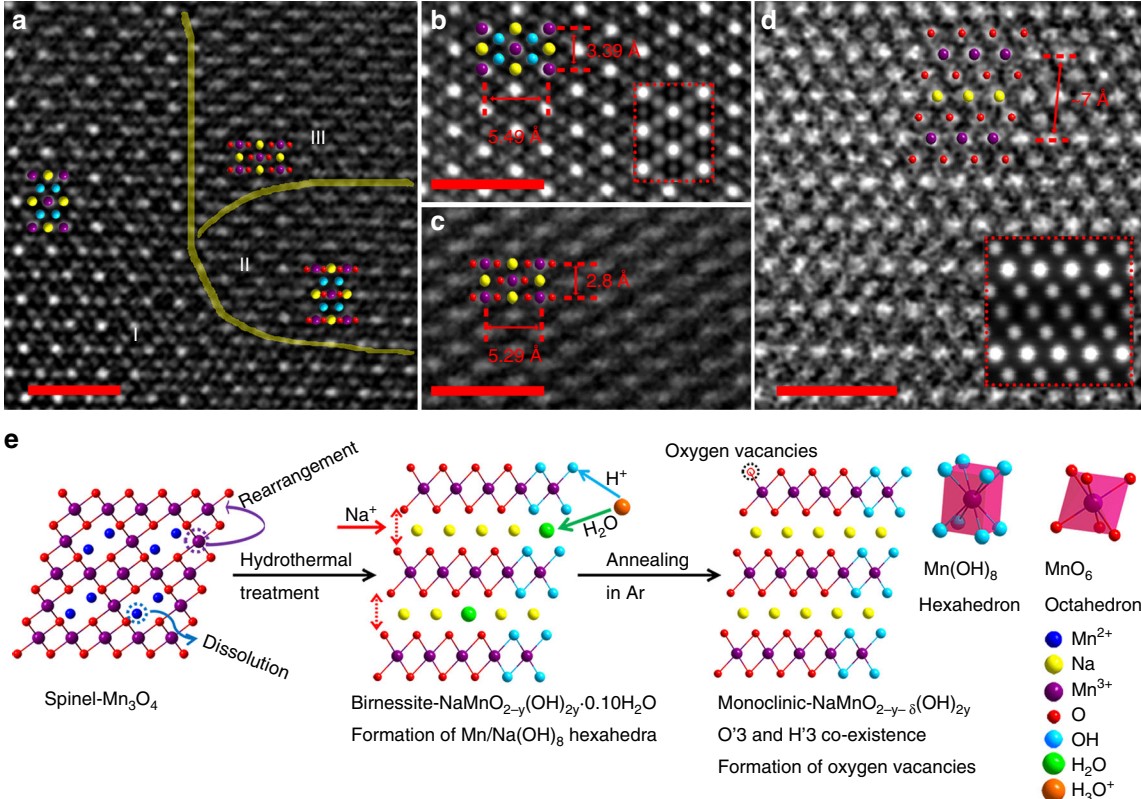

**Fig. 3** STEM images and formation mechanism. **a–c** HAADF-STEM images for the S4 sample along [001] zone axis; Scale bar, 1 nm (I, H′3 stacking zone; II, intermediate H′3/O′3 stacking zone; III, O′3 stacking zone). **d** HAADF-STEM image for the S4 sample along [010] zone axis; Scale bar, 1 nm. **e** Schematic illustration of the phase transition from spinel-$Mn_3O_4$ to monoclinic-$NaMnO_{2-y-\delta}(OH)_{2y}$. The simulated HAADF images along [001] and [010] zone axis are inserted in **b** and **d**, as outlined by red dotted frames, respectively

of the nanosheet, revealing the thickness of the nanosheet is about 5 nm.

**Crystal structure of monoclinic-$NaMnO_{2-y-\delta}(OH)_{2y}$ at atomic scale.** To further confirm the crystal structure of S4 at atomic scale, advanced spherical aberration-corrected electron microscope was utilized for detailed structural analysis using high-angle annular dark field (HAADF) and annular bright field (ABF) STEM. Fig. 3a–c show the HAADF-STEM images of the monoclinic-$NaMnO_{2-y-\delta}(OH)_{2y}$ nanosheet along the [001] zone axis. In the HAADF images, the Na, Mn, and O atomic arrangements can be clearly revealed, and the brighter dots represent Mn with the largest atomic number[18]. The major atom stacking agrees well with the structure of monoclinic-O′3 while domains with new stacking have been detected in Fig. 3a. Domains with two different stackings are zoomed in and shown in Fig. 3b and c, respectively, for comparison. Different from O′3 stacking (Fig. 3c), Fig. 3b presents H′3 stacking with $Mn(OH)_8$ and $Na(OH)_8$ hexahedra instead of $MnO_6$ and $NaO_6$ octahedra. The atomic arrangement shown in the HAADF image agrees well with the simulated result inserted in Fig. 3b. Although H atom cannot be observed from HAADF or ABF images, the O in hexahedra must bond with one H. Otherwise the valence of O should be −1, which is not stable for this structure. In the XPS results (Supplementary Fig. 4), −OH can still be detected from the O 1s spectra for the S4 sample after annealing, further supporting this speculation. Three different regions can be distinguished in Fig. 3a with zone I for H′3, zone III for O′3, and zone II for the intermediate transition state from H′3 to O′3. Compared to zone I, the Mn atoms are elongated in zone II due to O atoms migration, and the O–Mn–O atomic columns are

clearly seen in Fig. 3c, illustrating the structure transition from H′3 to O′3. Interestingly, stacking faults along the *ab* plane can be observed due to two anti-phase boundaries in H′3 stacking (Supplementary Fig. 7), which is beneficial for forming a lower energy boundary and improving structural stability[19,20], agreeing well with the SAED pattern in Fig. 2p. Fig 3d shows the HAADF image of the S4 viewed along [010] orientation and the experimental atomic configuration is in good agreement with the simulated one (red square in Fig. 3d). The interlayer distance for (001) planes was measured to be about 7 Å for S4, which is very close to that (7.2 Å) of birnessite-$Na_xMnO_{2-y}(OH)_{2-y} \cdot nH_2O$. Single $Mn(OH)_8$ hexahedron and $MnO_6$ octahedron are displayed in Fig. 3, and the corresponding crystal structures of H′3 and O′3 viewed along [001] and [010] orientations are shown in Supplementary Fig. 8, with Na atoms filled in. The ABF-STEM images along the [001] and [010] zone axes for H′3 and O′3 phases are shown in Supplementary Figs. 9 and 10, respectively, and corresponding lattice parameters are summarized and compared in Supplementary Table 2. The H′3 stacking with $Mn(OH)_8$ and $Na(OH)_8$ hexahedra is obviously sparser compared to normal O′3 stacking with $MnO_6$ and $NaO_6$ octahedra, resulting in bigger crystal unit cells and larger lattice parameters. Therefore, a larger *c* value is expected and demonstrated for S4, owing to the co-existence H′3 and O′3 phases with both hexahedra and octahedra. The crystallographic data of this H′3/O′3 monoclinic $NaMnO_{2-y-\delta}(OH)_{2y}$ are given in Supplementary Table 3. The weaker distortion of Mn-O distance and smaller β angle serve to suppress the Jahn–Teller distortion. From these observations, the as-obtained S4 sample was determined to be a special monoclinic-$NaMnO_{2-y-\delta}(OH)_{2y}$ with new polymorph consisting of both H′3 and O′3. The rearranged metal–oxygen (Mn–O) layers are not

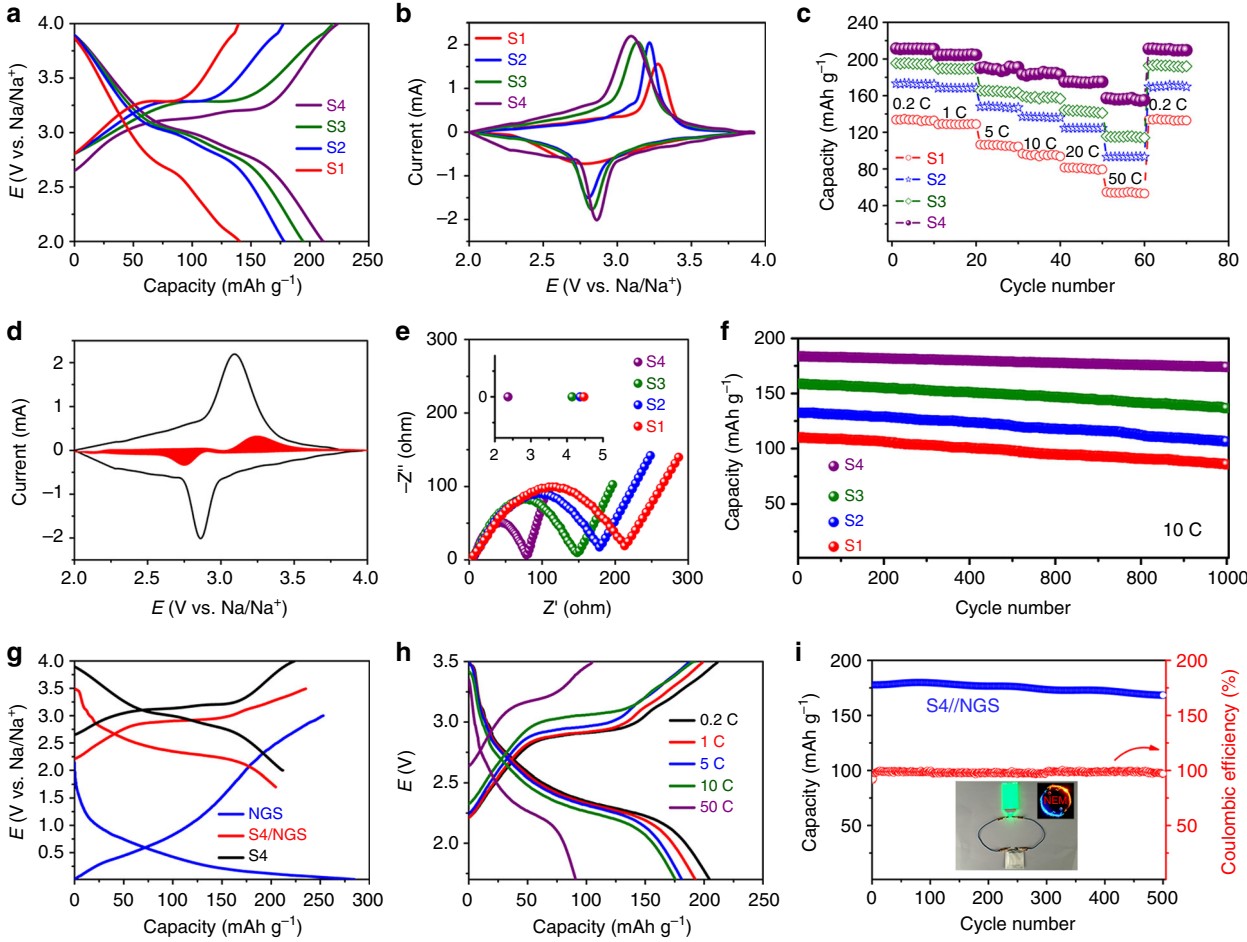

**Fig. 4** Sodium storage performance. **a** The first charge/discharge curves of S1–S4 electrodes. **b** CV curves of S1–S4 electrodes at a scan rate of 0.1 mV s$^{-1}$. **c** Rate capabilities of S1–S4 electrodes. **d** CV curve of the S4 electrode between 2.0 and 4.0 V with shadowed area representing the surface capacitive contribution at a scan rate of 0.1 mV s$^{-1}$. **e** EIS spectra of S1–S4 electrodes. **f** Cycle performances of S1–S4 electrodes. **g** Typical charge/discharge curves of the S4 cathode, the NGS anode, and the S4//NGS full cell. **h** Charge/discharge curves of the S4//NGS full cell at different C rates. The reversible capacity of the full cell at the high rate of 50 C is mainly limited by the NGS anode. **i** Cycle performance of the S4//NGS full cell at 10 C. Inset in (**i**) is the optical image of the punch cell device powering a LED

sufficiently close-packed with coexistence of two phases, which is different from that of normal monoclinic mentioned in the literature[7], probably resulting in the suppression of Jahn–Teller distortion from the framework of $MnO_6$ octahedra[21,22]. The formation of Mn–O–H bonds in hexahedra in S4 is speculated to be related to the "hydrothermal sodiation" process, which is illustrated in Fig. 3e. During the hydrothermal treatment, $Mn^{2+}$ in the $Mn_3O_4$ spinel tends to dissolve into the solution and is extracted from the lattice, which is similar to the $Mn^{2+}$ dissolution from $Mn_3O_4$ occurring in an electrochemical oxidation process[23]. The deinsertion of $Mn^{2+}$ gives rise to structure evolution with rearrangement of $Mn^{3+}$ and strong electrostatic repulsion between the O ions. To compensate the imbalanced charge, $H_3O^+$ and $Na^+$ insertion into the lattice occur, stabilizing the formed birnessite structure. After $H_3O^+$ intercalation, $H_2O$ molecules can stay at interlayer as crystal water while $H^+$ can interact with neighboring O layers, forming Mn–O–H bonds. In latter annealing process, the domains with Mn–O–H bonds evolve into the H′3 stacking while the rest Mn–O bonds of sample evolve into the O′3 stacking, forming this H′3/O′3 monoclinic.

**Sodium storage performance and full-cell device.** The sodium storage performances of S1–S4 samples in non-aqueous electrolyte were investigated using half cells. Fig. 4a shows the first charge/discharge curves of the S1–S4 electrodes at 0.2 C rate in the potential range of 2.0–4.0 V (vs. Na/Na$^+$). The initial charge/discharge capacities of the S1–S4 electrodes are about 139.0/140.6, 176.2/179.1, 218.8/194.9, and 223.9/211.9 mAh g$^{-1}$, respectively. The S4 electrode delivers the largest reversible capacity with a high Coulombic efficiency of about 94.6%. Note that the Coulombic efficiencies of the S1 and S2 electrodes are slightly larger than 100%, which is due to the extra Na$^+$ insertion from the electrolyte during discharge process. Compared with typical charge/discharge curves of O3-Na$_x$MnO$_2$ and P2-Na$_x$MnO$_2$, the charge/discharge curves of S1–S3 for birnessite-Na$_x$MnO$_{2-y}$(OH)$_{2y}$·nH$_2$O and S4 for monoclinic NaMnO$_{2-y-δ}$(OH)$_{2y}$ are very smooth and sloppy without multiple voltage plateaus, indicating suppressed structural change and phase transitions upon Na$^+$ extraction and insertion[10]. As illustrated in Fig. 4b, the cyclic voltammetry (CV) curves S1–S4 electrodes display a pair of non-sharp redox peaks between 2.6 and 3.2 V, corresponding to reversible intercalation and deintercalation of Na$^+$ into and from the layered structure. Importantly, the S4 electrodes shows the smallest potential difference between cathodic and anodic peaks, suggesting small polarization and fast electrode kinetics. To compare the rate performance, charge/discharge capacities of S1–S4 at different C rates as a function of

cycle number are shown in Fig. 4c. It is clear that the S4 electrode exhibits superior rate performance over S1–S3 electrodes, and can still deliver a large reversible capacity of 156.0 mAh g$^{-1}$ at a very high current rate of 50 C, outperforming the rate performance of previously reported O3-, P2-, and birnessite Na$_x$MnO$_2$·$n$H$_2$O (Table 1). To investigate the capacity contributions from surface pseudo-capacitance and ion intercalation, current contributions of surface-controlled process and diffusion-controlled process are separated by using Dunn's method[24]. The surface-controlled pseudo-capacitance contribution is represented by the red region in the CV curve at 0.1 mV s$^{-1}$ in Fig. 4d, indicating the surface pseudo-capacitive contribution is minor as compared to diffusion-controlled contribution. The diffusion controlled and surface capacitive contributions to the total stored charge of the S4 electrode at different scan rates are shown in Supplementary Fig. 11, clearly demonstrating the major capacity contribution for the NaMnO$_{2-y-\delta}$(OH)$_{2y}$ electrode is from bulk intercalation mechanism. Electrochemical impedance spectroscopy (EIS) measurements were carried out on half cells with the S1–S4 electrodes, and the obtained EIS spectra are shown in Fig. 4e. It is interesting to find that the charge transfer resistance (semi-circle) of the electrode decreases as the Na content increases and crystal water content decreases in the layered structure. Moreover, the inset in Fig. 4e shows the intercepts of the four EIS spectra with $Z'$-axis, demonstrating the S4 electrode has the smallest ohmic resistance. The greatly reduced ohmic resistance of the S4 electrode indicates improved electronic conductivity of the sample with introduction of O vacancies. DFT calculations have been carried out on monoclinic NaMnO$_2$ and NaMnO$_{2-\delta}$ with 4 at% O vacancies. It was found that the band gap of monoclinic NaMnO$_2$ reduces from 1.25 to 0.24 eV when 4 at% O vacancies are introduced into the lattice, indicating the enhancement of electronic conductivity of the material (Supplementary Fig. 12). Compared to S1–S3, the S4 sample possesses similar large interlayer distance but without crystal water blocking Na$^+$ diffusion, thus leading to accelerated Na$^+$ migration in the Na slabs. Therefore, the S4 electrode exhibits greatly improved rate performance as compared to S1–S3 and previously reported O3-Na$_x$MnO$_2$ and P2-Na$_x$MnO$_2$, benefiting from its fast electrode kinetics with both enhanced electron and ion transport. The cycle performances of S1–S4 electrodes are further compared in Fig. 4f with the plots of specific discharge capacity as a function of cycle number for 1000 cycles at 10 C. After 1000 charge/discharge cycles, the retained discharge capacities are 85.9, 106.3, 136.6, and 173.8 mAh g$^{-1}$ for S1–S4 electrodes, corresponding to 78.0%, 80.2%, 86.0%, and 94.6% capacity retentions, respectively. The charge/discharge curves of the S1–S4 electrodes (Supplementary Fig. 13a) display similar profiles as the initial charge/discharge curves. All the S1–S4 samples were further characterized by XRD and FESEM after 1000 cycles at 10 C. Supplementary Fig. 13b shows the XRD spectra of S1–S4 samples after 1000 charge/discharge cycles at 10 C, which agree well with the XRD spectra of the pristine S1–S4 samples. A new diffraction peak emerged at about 34° can be ascribed to Na$_2$CO$_3$, corresponding to the solid electrolyte interface (SEI) layer formed at the surface of the electrodes. Finally, the FESEM images of the S1–S4 samples after 1000 charge/discharge cycles demonstrate that the morphologies of all samples are well retained after cycling test, confirming the outstanding structural stability of the samples. As shown in Table 1, cycle performance and rate performance of the present S4 electrode are compared with those of reported layered Na$_x$MO$_2$ (M represents transition metal) in literature. Surprisingly, the S4 electrode possesses the best cycle performance and rate performance by far, indicating significantly improved structural stability and electrode kinetics. The excellent rate performance and cycle performance of the present S4 sample could be

attributed to its ultra-thin nanosheet morphology and the new polymorph. With the ultra-thin nanosheet morphology, the surface redox reactions can effectively shorten the ion diffusion paths and mitigate structural variation during charge/discharge[25]. As reported in literature[10], Aurbach and Choi et al. prepared Na-birnessite with similar nanosheet morphology. Although their samples have similar ultra-thin nanosheet morphology, the cycle performance and rate performance of their samples are much worse as compared to those of the present S4 sample. Therefore, in addition to the ultra-thin nanosheet morphology, the new polymorph of the S4 sample is effective to further improve its rate performance and cycle performance for Na ion storage. The volumetric capacity of the S4 powdery sample was calculated to be about 180 Ah L$^{-1}$. As compared in Supplementary Fig. 14 with other cathodes and anodes for LIBs, the S4 electrode presents relatively high volumetric capacity, indicating promising application for high energy SIBs. To demonstrate the future application of this monoclinic H′3/O′3 NaMnO$_{2-y-\delta}$(OH)$_{2y}$ cathode in SIBs, a pouch cell was constructed by using N-doped graphene nanosheets (NGS) as the anode (Supplementary Fig. 15). Fig. 4g shows the typical charge/discharge curves of the NaMnO$_{2-y-\delta}$(OH)$_{2y}$//Na half cell, NGS//Na half cell, and NaMnO$_{2-y-\delta}$(OH)$_{2y}$//NGS full cell at 0.2 C, respectively. As the average operating voltages of NaMnO$_{2-y-\delta}$(OH)$_{2y}$ and NGS are about 3.05 and 0.5 V, respectively, this full cell device delivers an average operating voltage of about 2.55 V. The initial discharge capacity of the full cell is 204.6 mAh g$^{-1}$ (based on the cathode material) in the voltage range of 1.7–3.5 V at 0.2 C rate. Even at a high rate of 10 C, the full cell can still deliver a high reversible capacity of about ~177 mAh g$^{-1}$ in Fig. 4h, displaying good rate performance. It is noted that the reversible capacity of the full cell is reduced to 91 mAh g$^{-1}$ at 50 C and the reversible capacity of the full cell at such a high rate is mainly limited by the NGS anode (Supplementary Fig. 16). Fig. 4i presents the cycle performance of the full cell cycled at 10 C for 500 continuous cycles. After 500 cycles, the full cell can still deliver a high reversible capacity of 168.4 mAh g$^{-1}$ with 95.1% capacity retention, demonstrating excellent cycling stability. After being charged at 19 mA (~10 C) to 3.5 V, a 2.5 × 2.5 cm$^2$ full cell device (based on the 2 × 2 cm$^2$ cathode and anode) can lighten a green light-emitting diode (LED; 2.5 V, 0.3 W) for 40 s.

**Understanding of superior Na storage performance**. To further understand the structural evolution in the monoclinic-NaMnO$_{2-y-\delta}$(OH)$_{2y}$ during Na extraction and insertion, in situ synchrotron XRD was carried out and the result is presented in Fig. 5a. During the charge and discharge processes, all diffraction peaks reveal continuous peak shift without peak splitting or new peak emergence, indicating solid solution reaction without phase transitions. This monoclinic NaMnO$_{2-y-\delta}$(OH)$_{2y}$ with solid solution reaction mechanism is completely different from previously reported O3-Na$_x$MnO$_2$ and P2-Na$_x$MnO$_2$ with multiple phase transitions during Na$^+$ intercalation/deintercalation. Accordingly, the variations of $c$-lattice parameter and unit cell volume as a function of charge and discharge potentials are calculated and shown in Fig. 5b and c. The $c$-lattice parameter changes slightly between 6.96 and 6.90 Å during charge and discharge, and the material only shows ~ 2% volume change between full sodiation and desodiation states. To further confirm this small volume change, we further did the ex-situ XRD and STEM measurements on the sample at full sodiation (2.0 V vs. Na/Na$^+$) and desodiation (4.0 V vs. Na/Na$^+$) states (Supplementary Figs. 17 and Fig. 18). All experimental results present the small lattice parameter change between full sodiation and desodiation states, confirming the small volume change for this ultrastable layered

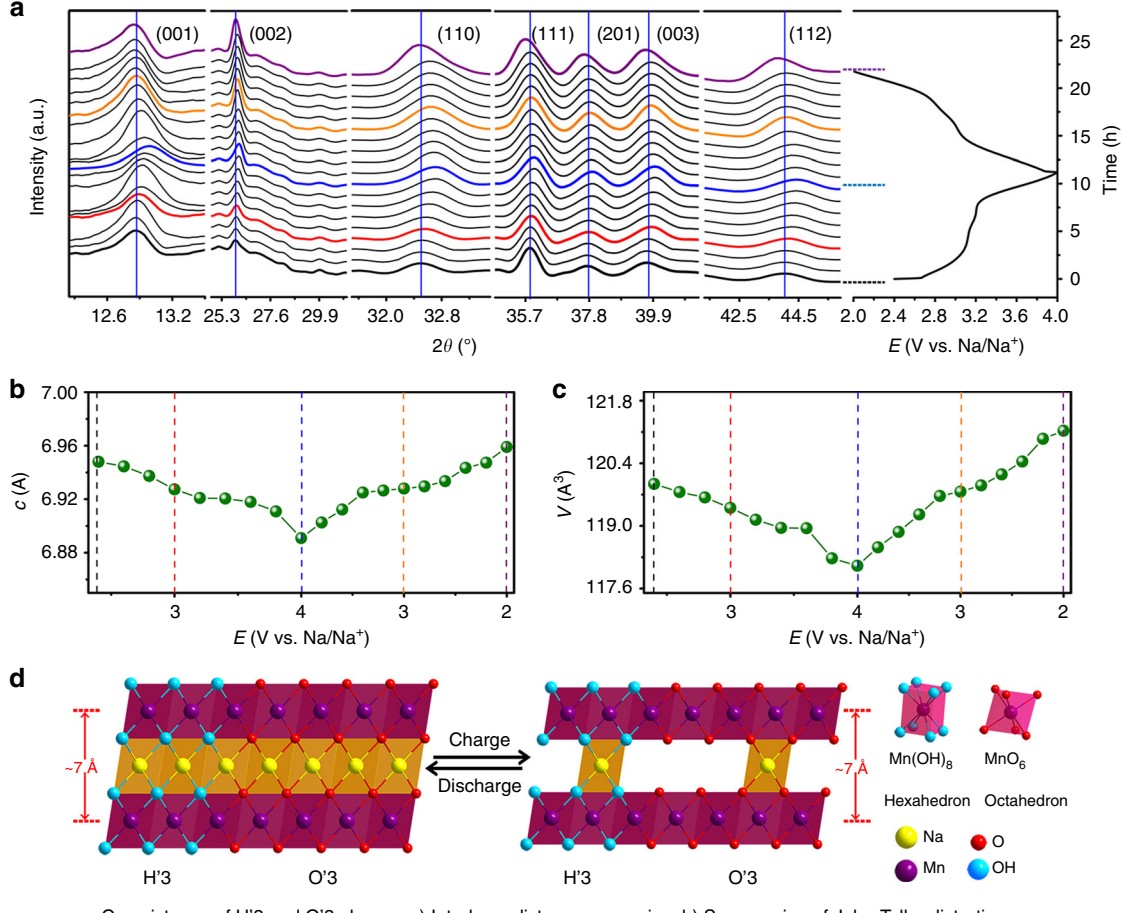

Co-existence of H'3 and O'3 phases: a) Interlayer distance expansion; b) Suppression of Jahn-Teller distortion
c) Fast ion and electron transport with oxygen vacancies; d) Non-phase transition.

**Fig. 5** Structure evolution upon Na intercalation/deintercalation. **a** In situ XRD patterns collected during the first discharge/charge of the Na/S4 half cell under a current rate of C/10 at a voltage range between 2.0 and 4.0 V. **b** and **c** Evolution of the $c$-lattice parameter and volume during charge/discharge processes. **d** Crystal structure evolution of the monoclinic H'3/O'3 $NaMnO_{2-y-\delta}(OH)_{2y}$ during charge/discharge processes

structure. Moreover, the surface and subsurface redox reactions of monoclinic $NaMnO_{2-y-\delta}(OH)_{2y}$ sample with nanoscale morphology could also facilitate small volume change during charge/discharge processes. As reported in literature[10], birnessite-$Na_{0.71}MnO_2 \cdot 0.25H_2O$ with a lower Na content exhibits an increase of $c$-lattice parameter at charged state, which is ascribed to the enhanced electrostatic repulsion between the negatively charged $MnO_2$ layers. In the present study, the slightly decreased $c$-lattice parameter could be ascribed to the reduced average charge on the oxygen ions when the Na concentration is significantly decreased, as observed for $LiCoO_2$[26]. Nevertheless, the monoclinic $NaMnO_{2-y-\delta}(OH)_{2y}$ with such a small volume change during charge/discharge represents a ultrastable layered structure with greatly suppressed Jahn–Teller distortion, thus resulting in outstanding cycling stability.

Compared to O3-, P2-, and monoclinic $Na_xMnO_2$, the monoclinic $NaMnO_{2-y-\delta}(OH)_{2y}$ with new polymorph possesses much larger interlayer distance of about 7 Å, which could greatly reduce $Na^+$ ion diffusion barrier in the Na slab between two $MnO_2$ layers. The expanded interlayer distance of the monoclinic $NaMnO_{2-y-\delta}(OH)_{2y}$ can be ascribed to the coexistence of H'3 phase with eight-coordinate hexahedral sites and O'3 phase with six-coordinate octahedral sites. With such a new stacking, the monoclinic structure retains the large interlayer distance as birnessite, even when the crystal water is completely removed from the interlayer. The H'3 stacking only forms via a

hydrothermal conversion from spinel $Mn_3O_4$ to birnessite. The directly synthesized birnessite does not have the H'3 stacking and interlayer distance will be significantly reduced when the crystal water is removed from the lattice. Therefore, the monoclinic $NaMnO_{2-y-\delta}(OH)_{2y}$ with O vacancies represent an ideal layered structure for sodium storage. First, the expanded interlayer without crystal water provides fast $Na^+$ migration channels and the O vacancies enable fast electron transport within this material, which together lead to fast electrode kinetics and excellent rate performance. Second, the suppressed Jahn–Teller distortion and minimal volume change during sodiation/desodiation result in superior structural stability and outstanding cycle performance. It should be noted that surface redox reactions on the nanosheets also contribute to the greatly improved rate performance and cycling stability for the electrode. Whether the outstanding Na storage performance of the present monoclinic $NaMnO_{2-y-\delta}(OH)_{2y}$ nanowall arrays is mainly due to the unique polymorph or the nanosheet morphology will be investigated in the future work.

## Discussion

In summary, a monoclinic $NaMnO_{2-y-\delta}(OH)_{2y}$ with new polymorph consisting of H'3 and O'3 stackings has been prepared by a hydrothermal method. The H'3 stacking with eight-coordinate hexahedral sites was first time reported for the layered structure

in the present work and it plays an important role in expanding the interlayer distance for the layered structure. With H′3 stacking, the monoclinic $NaMnO_{2-y-\delta}(OH)_{2y}$ without crystal water exhibits an expanded interlayer distance of about 7 Å, which is much larger than those of O3, P2 stackings. The expanded interlayer not only facilitates fast $Na^+$ diffusion in the Na slab but also leads to greatly improved structural stability by solid solution reaction mechanism without phase transitions. With further introduction of O vacancies, the as-prepared $NaMnO_{2-y-\delta}(OH)_{2y}$ nanowall arrays exhibit large specific reversible capacity of 211.9 mAh g$^{-1}$, excellent rate capability (156.0 mAh g$^{-1}$ at 50 C), and ultrastable cycle performance (94.6% capacity retention after 1000 cycles), making them promising cathode candidate for practical application in SIBs. Importantly, the present study demonstrates successful example and provides important insights in layered structure design for improved metal ion storage.

## Methods

**Materials synthesis**. The porous $Mn_3O_4$ nanowall arrays were deposited on the carbon cloth substrates by cathodic deposition. For the cathodic deposition, a three-electrode electrochemical cell was employed with a solution of 0.015 M manganese acetate (99%, Aldrich) and 0.015 M sodium sulfate (99%, Aldrich) as the electrolyte. The electrochemical cell consists of a carbon cloth (about 2 cm × 2 cm) as the working electrode, a platinum foil (about 4 cm$^2$) as the counter electrode, and a Ag/AgCl (in saturated KCl) as the reference electrode. The cathodic depositions were carried out on carbon cloth substrates at a constant potential of −1.6 V vs. Ag/AgCl for 600 s. After deposition, the carbon cloth supported Mn (OH)$_2$ was cleaned with distilled water and dried in the air at room temperature overnight to convert into $Mn_3O_4$ nanowall arrays. The carbon cloth substrate supported $Mn_3O_4$ nanowall arrays were immersed into a 40 mL 0.1 M NaOH solution and transferred into a 50 mL Teflon-lined stainless steel autoclave and heated at 160 and 180 °C for 10 h, respectively, to obtain the S1 and S2 samples. Differently, the carbon cloth substrate supported $Mn_3O_4$ nanowall arrays were first hydrothermally treated in a 50 mL 0.15 M NaOH solution at 80 °C for 8 h and subsequently transferred into a 50 mL 0.1 M NaOH solution at 180 °C for 8 h to obtain the S3 sample. The S4 sample was obtained by annealing the S3 sample at 300 °C in the Ar atmosphere for 12 h. The N-doped graphene nanosheets were prepared as the anode material according to a previous work[27].

**Materials characterization**. The Na/Mn atomic ratios and the crystal water contents of S1–S4 samples were determined by ICP and TGA measurements. The crystallographic information and phase purity of the samples were characterized by XRD (Bruker-AXS D8 Advance), Raman spectroscopy (Jobin–Yvon T6400 Micro-Raman system), and XPS (Phi Quantera SXM spectrometer using Al Kα X-ray as the excitation source). The morphology and microstructure of the samples were investigated by FESEM (Hitachi S4300), TEM, and HRTEM (FEI Tecnai 20). A JEM-ARM200F STEM fitted with a double aberration-corrector for both probe-forming and imaging lenses was used to perform HAADF and ABF imaging, which was operated at 200 kV. The attainable resolution of the probe defined by the objective pre-field is 78 pm. Simulation of HAADF images were performed by multislice method with the condition in the following: thickness: 80 nm; collection angle: 90–370 mrad; probe size: 0.78 nm; defocus: 25 Å.

**Electrochemical measurements**. The S1–S4 samples without any binders or conductive additives were directly used as the cathodes. Half cells using sodium foils as both counter and reference electrodes were assembled in Swagelok cells in a glove box for electrochemical measurements. For all electrochemical measurements, 1 M NaClO$_4$ in ethylene carbonate (EC) and propylene carbonate (PC) (v/v = 1:1) solution was used as the electrolyte and glass fibre was used as the separator. CV measurements were performed in the voltage range between 2 and 4 V (vs. Na/Na$^+$) at a scan rate of 0.1 mV s$^{-1}$ using CHI660D electrochemical workstation. Galvanostatic charge/discharge measurements were carried out in the voltage range between 2 and 4 V at different C rates (1 C = 243 mAh g$^{-1}$) using LAND CT2001A battery tester. To construct the full cell, the carbon cloth coated NGS was used as the anode to couple with the S4 cathode. The full cell was sealed inside polyethylene film. The mass ratio of S4/NGS was set to be 0.8 for the full cell to attain matched capacities for cathode and anode. The full cell was charged and discharged between 1.7 and 3.5 V at different C rates (1 C = 243 mAh g$^{-1}$). The calculation of the specific capacities for both half cells and full cells were based on the mass of cathode. The mass loadings of active materials on different substrates were measured by a Sartorius Analytical balance (CPA225D, with a resolution of 10 μg). The mass loadings of S1–S4 electrodes were in the range of 2–3 mg cm$^{-2}$. For the anode, NGS, conductive carbon black (Super P), and poly(vinylidenefluoride) (PVDF) with a mass ratio of 8:1:1 were mixed and stirred to form a homogeneous slurry, and then the slurry was coated onto carbon cloth with mass loading of 2–3 mg cm$^{-2}$.

**DFT calculations**. First-principles calculations were conducted based on the spin-polarized GGA+$U$[28,29] using the Perdew–Burke–Ernzerhof exchange and correlation functionals[30]. A plane-wave basis set and the projector-augmented wave (PAW) method[31,32], which is implemented in the Vienna ab initio simulation package (VASP)[28], was adopted. A gamma point mesh is performed with 5 × 9 × 5 k-points for NaMnO$_2$ and 2 × 3 × 5 k-points for the super cell of $NaMnO_{2-\delta}$ with 4 at% O vacancies. All the atoms in unit cells were fully relaxed to obtain the relaxed structures with a cutoff energy of 520 eV specified by the pseudopotentials on a plane-wave basis set. $U$-value of 3.9 eV was used for Mn[33].

## Data availability

Data that support the findings detailed in this study are available in the Supplementary Information and its Article or from the corresponding author upon reasonable request.

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

## Acknowledgements

This work was supported by National Natural Science Foundation of China (No. 51572129, 51772154), International S&T Cooperation Program of China (No. 2016YFE0111500), Natural Science Foundation of Jiangsu Province (No. BK20170036). J. K.S. and Y. S.Meng are grateful for the financial support from the funding by USA National Science Foundation under Award Number DMR1608968. J.K.S. and Y.S.M.'s computational work utilizes computing resources provided by Triton Shared Computing Cluster (TSCC) at the University of California, San Diego (UCSD), the National Energy Research Scientific Computing Center (NERSC), and the Extreme Science and Engineering Discovery Environment (XSEDE) supported by the National Science Foundation under Grant No. ACI-1053575

## Author contributions

The project was conceived by H.X.; X.H.Z. synthesized the samples and performed structural characterizations and electrochemical measurements; L.G., Q.H.Z., X.Y.L. and J.Z.L performed TEM experiments and structural analysis; Q.L. and S.L. performed in-situ synchrotron XRD and data analysis; T.T.C. performed ex-situ XRD and Raman measurements; J.K.S. and Y.S.M. performed DFT calculations; H.X., X.H.Z. and J.Z.L. co-wrote the manuscript; all authors analyzed the results and commented on the manuscript.

## Additional information

**Competing interests:** The authors declare no competing interests.

