## [Peer Review File · Nature Communications]

Reviewers' comments:

Reviewer #1 (Remarks to the Author):

The manuscript claimed the synthesis of monoclinic $\text{NaMnO}_2 \cdot y\text{-}\delta(\text{OH})_2$ with large interlayer spacing, and demonstrated extraordinarily good electrochemical properties, including a large specific capacity of 211.9 mAh g⁻¹, 94.6% capacity retention after 1000 cycles at 10C, and 156.0 mAh g⁻¹ at 50 C. The authors have characterized the structure and chemistry of the monoclinic $\text{NaMnO}_2 \cdot y\text{-}\delta(\text{OH})_2$ and ascribed the enhanced electrochemical performance to the new polymorph with ultrastable layered structure and fast Na⁺ migration.

However, the major arguments are not convincing at all. First, instead of the insertion/extraction of Na ions in the layered structure, the very fast electrochemical kinetics achieved on nanoscale composites with carbon fabric support in this work should be pseudo-capacitive in nature. In order to verify their argument, the authors should prepare powdery $\text{NaMnO}_2 \cdot y\text{-}\delta(\text{OH})_2$ active materials without conductive support and compare the corresponding electrochemical and structural properties with those of nanoscale composites. Second, it is hard for the reviewer to believe that only ~1% volume change occurs between full sodiation and desodiation states during the insertion/extraction storage processes. In addition to the in-situ XRD results, the authors are suggested to use first principles calculations to confirm the structural change theoretically. Third, it is not convincing to claim the existence of oxygen vacancies based on the XPS results, clear evidence of the oxygen species should be provided from e.g. XANES or EELS analysis. Therefore, in my opinion, the manuscript may be improved if these major problems could be carefully considered and clarified.

Reviewer #2 (Remarks to the Author):

The paper by Xia et al. reports a new polymorph based on NaMnO_2 as a cathode with ultrafast kinetics – up to 50C - and very stable electrochemical performance. They formed a Na-birnessite structure containing crystal water with large interlayer spacing and further removed the crystal water to maintain the large c-parameter. The newly formed compound has a H³/O³ stacking and allows very fast Na kinetics. The work is sound and well-presented but a few questions arise.

1. What is the effect of crystal water on the electrolyte degradation for birnessite-type samples S1-S3? Similar Na-birnessite compounds containing crystal water have been previously described (such as Li et al., *CrystEngComm*, 2016, 18, 3136), but the cycling performance were low: what is the authors explanation for this different behaviour?

2. What is the H³ and O³ ratio in the monoclinic compound? Is there an ideal ratio that would allow even higher Na kinetics? Is it more energetically favourable to extract/reinsert Na from the H³ or O³ stacking?

3. The authors claim that oxygen vacancies are introduced during annealing, the cooling method can also influence the presence of O²/Mn vacancies and thus further reduce the Jahn-Teller distortion. Has this been investigated?

4. The XRD is not convincing: sample S1 is clearly different from the others - some peaks are missing and some peaks are extra. The peaks ratio also varies depending on the sample. A proper refinement should be provided.

5. The SEM reveals highly porous materials, what is the BET of such compounds? What is the porosity and tortuosity? At 50C is it really an insertion mechanism or a surface-driven reaction?

6. I would suggest that the authors provide volumetric capacity values and compare their performance with other types of fast kinetics cathode materials to illustrate their improved performance. The key to the fast kinetics seems to rely on the large interlayer spacing, as already illustrated by Wan et al. on expanded graphite (*Nature Communications* volume 5, Article number: 4033 (2014)). A comparison

with anode materials with fast kinetics would also be helpful.

7. Can the authors provide the load curves at 50C in Figure 4g?

8. The authors claim a very stable compound, yet a post-mortem analysis (SEM and XRD) is lacking: the evolution of the structure, the load curves and the morphology after long term cycling is essential.

Reviewer #3 (Remarks to the Author):

This manuscript reports for the first time the electrochemical performances of sodium manganese hydroxide as Na ion electrode material. The authors report a capacity of 211.9mAh/g with also an outstanding rate capability. Such excellent performance is explained by the structure of the material. The manuscript is well written and clear and the large amount of experimental techniques as well as results deserve this paper to be published as main article. Please find below few comments:

Figure 1 and S1 : please use the same color between xrd patterns /TGA/and structure for Sx

line 96-97: not clear, please be more precise

line 107: it is not true, the xrd patterns are not similar at all, please change

Line126: "other solution methods" , please give more detail, add reference

lines 143-147: XPS results need more detail and explanation, it is not clear

line 151 and 173: figure 2b is FESEM image of S1 or Mn₃O₄ ? please correct this mistake

figure2: please reorganize by material instead of magnification to clarify

line 201 to 203, typo trouble with appearance of square for the number of zone

line 249-251, the text is not consistent with the figure 3c

figure 4d: please give nyquist plot orthonormal

Reviewer #1:

The manuscript claimed the synthesis of monoclinic $\text{NaMnO}_{2-y-\delta}(\text{OH})_{2y}$ with large interlayer spacing, and demonstrated extraordinarily good electrochemical properties, including a large specific capacity of 211.9 mAh g^{-1} , 94.6% capacity retention after 1000 cycles at 10C, and 156.0 mAh g^{-1} at 50 C. The authors have characterized the structure and chemistry of the monoclinic $\text{NaMnO}_{2-y-\delta}(\text{OH})_{2y}$ and ascribed the enhanced electrochemical performance to the new polymorph with ultrastable layered structure and fast Na^+ migration.

1) However, the major arguments are not convincing at all. First, instead of the insertion/extraction of Na ions in the layered structure, the very fast electrochemical kinetics achieved on nanoscale composites with carbon fabric support in this work should be pseudo-capacitive in nature. In order to verify their argument, the authors should prepare powdery $\text{NaMnO}_{2-y-\delta}(\text{OH})_{2y}$ active materials without conductive support and compare the corresponding electrochemical and structural properties with those of nanoscale composites.

Response: Thanks for the valuable comments. There is still some impurity in the directly synthesized powdery material. Alternatively, phase pure powdery $\text{NaMnO}_{2-y-\delta}(\text{OH})_{2y}$ active materials were obtained from the carbon fabric support by ultrasonication. 20 mg powders were collected for the S1, S2, S3, and S4 samples, respectively. For electrochemical measurements, active material, conductive carbon black (Super P), and poly(vinylidene fluoride) (PVDF) with a mass ratio of 75:15:10 were mixed and stirred to form a homogeneous slurry, and then the slurry was coated onto Al foil with mass loading of $\sim 3 \text{ mg cm}^{-2}$. As shown in Figure R1, the initial charge/discharge capacities of the powdery S1, S2, S3, and S4 electrodes are about 138.3/139.0, 174.6/177.7, 209.9/190.9, and 220.4/211.3 mAh g^{-1} , respectively, agreeing well with the nanowall array electrodes. To verify whether the fast electrode kinetics is due to pseudo-capacitance or intercalation, current contributions of surface-controlled process and diffusion-controlled process are separated by using Dunn's method (Brezesinski, T., et al. Nat. Mater. 2010, 9, 146-151). The surface-controlled pseudo-capacitance contribution is represented by the red region in the CV curve at 0.1 mV s^{-1} in Figure R1c, indicating the surface pseudo-capacitive contribution is minor as compared to diffusion-controlled contribution. The diffusion-controlled and surface capacitive contributions to the total stored charge of the S4 electrode at different scan rates are shown in Figure R1d, clearly demonstrating

the major capacity contribution for the $\text{NaMnO}_{2-y-\delta}(\text{OH})_{2y}$ electrode is from bulk intercalation mechanism. Therefore, the enhanced electrode kinetics of the present $\text{NaMnO}_{2-y-\delta}(\text{OH})_{2y}$ nanowall array electrode can be attributed to its unique polymorph with expanded interlayer distance, favoring fast Na^+ ion intercalation/deintercalation. Related discussion and Figures have been added in the revised manuscript and Supplementary Information.

Figure R1. (a) The first charge/discharge curves of the powdery S1-S4 electrodes at 0.2 C. (b) CV curves of the S4 electrode at different scan rates of 0.1-1.0 mV s^{-1} . (c) CV curve of the S4 electrode between 2.0 and 4.0 V with shadowed area representing the surface capacitive contribution at a scan rate of 0.1 mV s^{-1} . (d) Diffusion-controlled capacity contributions at different scan rates for the S4 electrode.

2) Second, it is hard for the reviewer to believe that only $\sim 1\%$ volume change occurs between full sodiation and desodiation states during the insertion/extraction storage processes. In addition to the in-situ XRD results, the authors are suggested to use first principles calculations to confirm the structural change theoretically.

Response: Thanks for the valuable comments. We are sorry that we made a mistake when calculating the volume change between full sodiation and desodiation states. $\sim 1\%$ volume change was calculated between the as-prepared state and full desodiation state.

Because the as-prepared state (~ 2.5 V) is a little different from the full sodiation state (2 V), a small error was introduced in the volume change calculation. We recalculated the volume change between full sodiation and desodiation states (2-4 V), and the value was determined to be about 2%. To confirm this small volume change, we further did the ex-situ XRD (Figure R2) and STEM (Figure R3) measurements on the sample at full sodiation and desodiation states. All experimental results present the same lattice parameter change between full sodiation and desodiation states, confirming the small volume change. Due to two phase coexistence in this sample, we are not able to construct the unit cell for the first principle calculation. In addition, the H occupancy in this structure is still not clear, and theoretical calculation without precise structural model could not reflect the real structural change between sodiation and desodiation. We believe the coherent in-situ XRD, ex-situ XRD, and STEM results can actually reflect the small volume change of this material during sodiation/desodiation process. Figure R2 and R3 have been added in the Supporting Information in the revised manuscript.

Figure R2. Ex-situ XRD patterns collected during the first discharge/charge of the Na/S4 half cell under a current rate of C/10 in a voltage range between 2.0 and 4.0 V.

Figure R3. (a,b) HAADF-STEM images for the full sodiation sample along [001] zone axis; Scale bar, 1 nm (a, H'3 stacking zone; b, O'3 stacking zone). (c,d) HAADF-STEM images for the full desodiation state of S4 sample along [001] zone axis; Scale bar, 1 nm (c, H'3 stacking zone; d, O'3 stacking zone). (e) HAADF-STEM image for the full sodiation state of S4 sample along [010] zone axis. (f) HAADF-STEM image for the full desodiation state of S4 sample along [010] zone axis.

3) Third, it is not convincing to claim the existence of oxygen vacancies based on the XPS results, clear evidence of the oxygen species should be provided from e.g. XANES or EELS analysis. Therefore, in my opinion, the manuscript may be improved if these major problems could be carefully considered and clarified.

Response: Thanks for the valuable comments. To confirm the existence of oxygen vacancies, we further carried out XANES measurements on the S3 and S4 samples. Figure R4 shows the XANES spectra of Mn L_{II,III}-edges of S3 and S4 (Standard XANES spectra of Mn L_{II,III}-edges for MnO and Mn₂O₃ are also included for reference). It is clear that the Mn L_{II,III}-edge shifts to lower energy for the S4 as

compared to that of S3, demonstrating decrease of average Mn valence and existence of oxygen vacancies in S4. Combining the XANES and XPS results, we can conclude the existence of oxygen vacancies in the S4 sample. XANES results and related discussion have been added in the revised manuscript and Supporting Information.

Figure R4. Mn $L_{II,III}$ -edge XANES spectra measured for the S3 and S4 samples.

Reviewer #2:

The paper by Xia et al. reports a new polymorph based on NaMnO_2 as a cathode with ultrafast kinetics – up to 50C - and very stable electrochemical performance. They formed a Na-birnessite structure containing crystal water with large interlayer spacing and further removed the crystal water to maintain the large c-parameter. The newly formed compound has a H^3/O^3 stacking and allows very fast Na kinetics. The work is sound and well-presented but a few questions arise.

1. What is the effect of crystal water on the electrolyte degradation for birnessite-type samples S1-S3? Similar Na-birnessite compounds containing crystal water have been previously described (such as Li et al., CrystEngComm, 2016, 18, 3136), but the cycling performance were low: what is the authors explanation for this different behaviour?

Response: Thanks for the good questions. For Na-birnessite, the crystal water could be extracted from the lattice during charge at high voltage, which could lead to side reactions with electrolyte and increased instability of the birnessite structure. As

observed from Figure 4e, S1-S3 electrodes show an improvement in cycle performance (capacity retentions from 78.0% to 80.2% to 86.0% for 1000 cycles) as the content of crystal water decreases, which suggests the existence of crystal water in lattice could deteriorate the cycling stability. The difference between our S1-S3 and usual Na-birnessite is that our S1-S3 are obtained from Mn_3O_4 and possess the $\text{Mn}(\text{OH})_8$ and $\text{Na}(\text{OH})_8$ hexahedra in the structure. Even when the crystal water is removed from S1-S3, the layered structure is still stable due to H'3 stacking as demonstrated in the S4 sample. For the usual Na-birnessite, however, the layered structure could be greatly degraded or distorted due to severe interlayer distance change when crystal water is removed, thus resulting in much worse cycle performance as compared to our S1-S3 samples.

2. What is the H'3 and O'3 ratio in the monoclinic compound? Is there an ideal ratio that would allow even higher Na kinetics? Is it more energetically favourable to extract/reinsert Na from the H'3 or O'3 stacking?

Response: Thanks for the good questions. The H'3 to O'3 ratio can be estimated from the ratio of $\text{Mn}^{3+}\text{-OH}$ to $\text{Mn}^{3+}\text{-O}$ bonds, which can be evaluated from O 1s XPS spectrum in Supplementary Fig. 3. The H'3 to O'3 ratio is estimated to be about 1:4 for S4. By changing the synthesis conditions (hydrothermal temperature), we can also obtain the S4-1 and S4-2 samples with lower H'3 to O'3 ratios of 1:5 and 1:4.6, respectively (Figure R5). The S4, S4-1, and S4-2 electrodes can deliver reversible capacities of 156, 100, and 130 mAh g^{-1} at 50 C, respectively, indicating that a higher H'3 content favors faster Na^+ transport kinetics. However, in our present work, the highest H'3 to O'3 ratio can be obtained in the S4 sample. Without H'3, the O'3 stacking only affords small interlayer spacing (small *c*-lattice parameter), which is not favorable for Na^+ diffusion. Therefore, in theory, it is more energetically favourable to extract/insert Na^+ from/into the H'3 stacking with large interlayer spacing.

Figure R5. (a) O1s core-level XPS spectra of S4, S4-1, and S4-2 samples, respectively. (b) The charge/discharge curves of the S4, S4-1, and S4-2 electrodes at 50 C.

3. The authors claim that oxygen vacancies are introduced during annealing, the cooling method can also influence the presence of O₂/Mn vacancies and thus further reduce the Jahn-Teller distortion. Has this been investigated?

Response: Thanks for the good comments. In the present study, the oxygen vacancies are introduced in the material to further improve its electron transport. We appreciate the reviewer's good suggestion that the cooling method could also influence the O/Mn vacancies and further reduce Jahn-Teller distortion. However, this study was not involved in the present work and we will investigate the cooling effect on its structural and electrochemical properties in the future.

4. The XRD is not convincing: sample S1 is clearly different from the others - some peaks are missing and some peaks are extra. The peaks ratio also varies depending on the sample. A proper refinement should be provided.

Response: Thanks for the comments. We are sorry that the XRD of S1 is not consistent with those of other samples. The XRD patterns for all samples have been measured again carefully and the consistent results are shown in Figure R6. To do the proper refinement, the powdery S1 sample is obtained from the carbon fabric support by ultrasonication for XRD. The refinement result is shown in Figure R7.

Figure R6. XRD patterns of the Mn_3O_4 and S1-S4 samples.

Figure R7. Refinement pattern of the XRD data for the powdery S1 sample, $R_{wp} = 8.99\%$.

5. The SEM reveals highly porous materials, what is the BET of such compounds? What is the porosity and tortuosity? At 50 C is it really an insertion mechanism or a surface-driven reaction?

Response: Thanks for the valuable comments. The nitrogen adsorption-desorption isotherms of the S4 powdery sample were measured and shown in Figure R8a with inserted pore size distribution curve, indicating typical mesoporous structure. The pore size distribution curve presents a relatively narrow pore size distribution around 9 nm. According to the BET computational method, the specific surface area of the S4 sample is about $105.3 \text{ m}^2 \text{ g}^{-1}$. Although the sample has large surface area, the major capacity contribution is from diffusion-controlled process as evaluated from the current separation method using CV (Figure 4d and Supplementary Fig. 11 in the revised manuscript). Therefore, even at a high rate of 50 C, the major capacity contribution is based on the insertion mechanism instead of surface-driven reaction. As shown in Figure R8b, the charge/discharge curves of the S4 electrode at 50 C still present well-defined voltage plateaus but not the straight lines for surface-driven

reaction. Related discussion on the charge storage mechanism of the porous electrode has been added in the revised manuscript.

Figure R8. (a) Nitrogen adsorption-desorption isotherms of the powdery S4 sample (Inset is the pore-size distribution of the powdery S4 sample). (b) Charge/discharge curves of the S4 electrode at 0.2 C and 50 C.

6. I would suggest that the authors provide volumetric capacity values and compare their performance with other types of fast kinetics cathode materials to illustrate their improved performance. The key to the fast kinetics seems to rely on the large interlayer spacing, as already illustrated by Wan et al. on expanded graphite (Nature Communications volume 5, Article number: 4033 (2014)). A comparison with anode materials with fast kinetics would also be helpful.

Response: Thanks for the valuable comments. The paper on expanded graphite (Nature Comm. 5, 2014, 4033) has been cited in this work. The volumetric capacity of the S4 sample was calculated to be about 180 Ah l⁻¹. As compared in Figure R9 with other cathodes and anodes for lithium-ion batteries, the S4 electrode presents relatively high volumetric capacity, indicating promising application for high energy sodium-ion batteries. Figure S9 has been added in the Supporting Information in the revised manuscript.

Figure R9. The specific volumetric capacities of the S4 sample and other electrode materials in literature.

7. Can the authors provide the load curves at 50C in Figure 4g?

Response: Thanks for the valuable comments. The charge/discharge curves of the full cell at 50 C has been added in Figure 4g in the revised manuscript. The reversible capacity of the full cell at high rate is mainly limited by the NGS anode.

Figure R10. Typical charge/discharge curves of the S4 cathode, the NGS anode, and the S4/NGS full cell at 50 C.

8. The authors claim a very stable compound, yet a post-mortem analysis (SEM and XRD) is lacking: the evolution of the structure, the load curves and the morphology after long term cycling is essential.

Response: Thanks for the valuable comments. To do the post-mortem analysis, all the S1-S4 samples were further characterized after 1000 cycles at 10 C. The charge/discharge curves of the S1-S4 electrodes (Figure R11a) display similar profiles as the initial charge/discharge curves. Figure R11b shows the XRD spectra of S1-S4 samples after 1000 charge/discharge cycles at 10 C, which agree well with the XRD spectra of the pristine S1-S4 samples. A new diffraction peak emerged at about 34° can be ascribed to Na_2CO_3 , corresponding to the SEI layer formed at the surface of the electrodes. Finally, the FESEM images of the S1-S4 samples after 1000 charge/discharge cycles demonstrate that the morphologies of all samples are well retained after cycling test, confirming the outstanding structural stability of the samples. Figure R11 has been added in the Supporting Information and related discussion has been added in the revised manuscript.

Figure R11. (a) The 1000th charge/discharge curves of the S1-S4 electrodes at 10 C. (b) XRD patterns of the S1-S4 samples after 1000 cycles at 10 C. (c,g) FESEM images of the S1 sample after 1000 cycles at 10 C. (d,h) FESEM images of the S2 sample after 1000 cycles at 10 C. (e,i) FESEM images of the S3 sample after 1000 cycles at 10 C. (f,j) FESEM images of the S4 sample after 1000 cycles at 10 C. (c-f) Scale bar, $50 \mu\text{m}$. (g-j) Scale bar, $2 \mu\text{m}$.

Reviewer #3

This manuscript reports for the first time the electrochemical performances of sodium manganese hydroxide as Na ion electrode material. The authors report a capacity of 211.9mAh/g with also an outstanding rate capability. Such excellent performance is explained by the structure of the material. The manuscript is well written and clear and the large amount of experimental techniques as well as results deserve this paper to be publish as main article. Please find below few comments:

Figure 1 and S1 : please use the same color between xrd patterns /TGA/and structure for Sx

Response: Thanks for the good suggestion. We have used the same color for S1-S4 sample in Figure 1 and Figure S1 in the revised manuscript.

line 96-97: not clear, please be more precise

Response: Thanks for the good comments. The sentence “Importantly, the high Na content birnessite- $\text{NaMnO}_{2-y}(\text{OH})_{2y}\cdot 0.10\text{H}_2\text{O}$ was first time prepared via the two-step “hydrothermal sodiation”.” has been changed as “Importantly, birnessite- $\text{Na}_x\text{MnO}_{2-y}(\text{OH})_{2y}\cdot 0.10\text{H}_2\text{O}$ with a high Na content of $x=1$ was first time prepared via the two-step “hydrothermal sodiation”.”.

line 107: it is not true, the xrd pattern are not similar at all, please change

Response: Thanks for the good comments. The XRD patterns of the samples were carefully measured again and the consistent result has been provided in Figure 1b in the revised manuscript.

Figure R6. XRD patterns of the Mn_3O_4 and S1-S4 samples.

Line126: "other solution methods" , please give more detail, add reference

Response: Thanks for the good comments. “other solution methods” has been changed as “other solution methods using Mn²⁺ or Mn⁷⁺ precursors”, and related reference (Nam, et al. *Chem. Mater.* 2015, 27, 3721-3725) has been added in the revised manuscript.

lines 143-147: XPS results need more detail and explanation, it is not clear

Response: Thanks for the good comments. More detail and explanation has been added for the XPS results to provide clear elucidation of valence change of Mn and introduction of oxygen vacancies for S4 sample.

X-ray photoelectron spectroscopy (XPS) results further confirm the Mn oxidation state change in Mn₃O₄ and S1-S4 samples (Supplementary Fig. 3). The energy difference between the two peaks (or peak separation) in the Mn 3s core-level XPS spectra is correlated to the average Mn valence. From S1 to S4, the peak separation keeps increasing, indicating the average Mn valence is decreasing. Different peaks can be detected from the Mn 2p core-level XPS spectra of S1-S4, where peaks located at 642.3, 641.1 and 640.2 eV correspond to Mn⁴⁺, Mn³⁺, Mn²⁺, respectively. The emergence of Mn²⁺ in S4 sample further confirms the existence of oxygen vacancies after heat treatment in the Ar atmosphere. As shown in the O 1s core-level XPS spectra, a component corresponding to Mn³⁺-O-H species can be found for all S1-S4 samples. No crystal water exists in the S4 sample as confirmed by the TGA result, suggesting the existence of –OH bonds in S4 even without crystal water in the lattice.

line 151 and 173: figure 2b is FESEM image of S1 or Mn₃O₄ ? please correct this mistake

Response: We are sorry for the mistake. Figure 2b shows FESEM image of Mn₃O₄ and the mistake has been corrected in the revised manuscript.

figure2: please reorganize by material instead of magnification to clarify

Response: Thanks for the good suggestion. Figure 2 has been reorganized by material in the revised manuscript.

line 201 to 203, typo trouble with appearance of square for the number of zone

Response: Thanks for the correction. The typo error has been corrected in the revised manuscript.

line 249-251, the text is not consistent with the figure 3c

Response: Thanks for the valuable comments. The text has been modified as “In the latter annealing process, the domains with Mn-O-H bonds evolve into the H’3 stacking while the rest Mn-O bonds of sample evolve into the O’3 stacking, forming this H’3/O’3 monoclinic”.

figure 4d: please give nyquist plot orthonormal

Response: Thanks for the valuable comments. The orthonormal Nyquist plot has been provided in Figure 4d in the revised manuscript.

Figure R12. Nyquist plots of the S1-S4 electrodes.

Reviewers' comments:

Reviewer #1 (Remarks to the Author):

The reviewer has carefully read the response and the revised manuscript and SI. However, it is found that the raised major arguments were not satisfactorily addressed by the authors.

1) In order to ascertain their claim that the achieved surprising rate capability is due to the monoclinic $\text{NaMnO}_{2-y-\delta}(\text{OH})_2$ with new polymorph, instead of the favorable nanostructure (ultra-thin nanowall arrays on carbon fabric), the authors should prepare powdery active materials and run the tests again. The so called powdery materials obtained from ultrasonication are still ultra-thin nanowall structure. In this context, the superior rate capability is achieved by effectively shortening the diffusion distance of Na ions in the nanowall structure. Similar phenomenon about enhanced kinetics of ultra-thin silicate materials with poor intrinsic electronic/ionic conductivity has been reported by Itaru Honma et. al. (Nano Lett. 2012, 12, 1146–1151). They also found that, owing to the shallow insertion of Li into the ultra-thin material, structural integrity of these fragile materials were substantially enhanced during electrochemical cycling. Therefore, it is not persuading for the authors to claim that the ultrafast and ultrastable Na ion storage is induced by the newly designed crystal structure.

2) Again, surface and subsurface redox reactions (pseudo-capacitive processes) may explain the small volume change occurred upon charging/discharging in this work.

3) To confirm the existence of oxygen vacancies, in addition to the Mn L-edge spectrum, the O K-edge spectrum should be provided and the variations between the corresponding O pre edge and O K-edge main peak should be carefully compared and discussed.

Reviewer #2 (Remarks to the Author):

The authors positively answered to the questions raised. It is very nice to observe a retained morphology even after 1000 cycles at fast rate. I am still skeptical however about the quantification they provided using these XPS data. I also find it doubtful that the SEI can be observed by XRD, that would mean a very thick SEI, which is not apparent on the post mortem SEM images. Despite these minor points, I recommend publication of the manuscript as all the points have been answered properly.

Reviewer #1:

The reviewer has carefully read the response and the revised manuscript and SI. However, it is found that the raised major arguments were not satisfactorily addressed by the authors.

1) In order to ascertain their claim that the achieved surprising rate capability is due to the monoclinic $\text{NaMnO}_{2-y-\delta}(\text{OH})_{2y}$ with new polymorph, instead of the favorable nanostructure (ultra-thin nanowall arrays on carbon fabric), the authors should prepare powdery active materials and run the tests again. The so called powdery materials obtained from ultrasonication are still ultra-thin nanowall structure. In this context, the superior rate capability is achieved by effectively shortening the diffusion distance of Na ions in the nanowall structure. Similar phenomenon about enhanced kinetics of ultra-thin silicate materials with poor intrinsic electronic/ionic conductivity has been reported by Itaru Honma et. al. (Nano Lett. 2012, 12, 1146–1151). They also found that, owing to the shallow insertion of Li into the ultra-thin material, structural integrity of these fragile materials were substantially enhanced during electrochemical cycling. Therefore, it is not persuading for the authors to claim that the ultrafast and ultrastable Na ion storage is induced by the newly designed crystal structure.

Response: We greatly appreciate reviewer's insightful comments. We agree with the reviewer that the excellent rate performance of the present electrode should be attributed to their nanoscale morphology, which effectively shortens the ion diffusion paths. We are sorry that we didn't highlight the nanostructure's effect on the electrochemical performance in the previous version of manuscript. The influence of the nanowall morphology on the electrochemical performance of the present $\text{NaMnO}_{2-y-\delta}(\text{OH})_{2y}$ electrode has been added in the revised manuscript.

On the other hand, due to the layered crystal structure, Birnessite Na_xMnO_2 always grows into the ultra-thin nanosheet morphology by various synthesis methods. We are not able to prepare bulk powdery samples with micro-sized particles for comparison. Nevertheless, to demonstrate the superior rate performance and cycling stability of the present $\text{NaMnO}_{2-y-\delta}(\text{OH})_{2y}$ electrode can also be ascribed to its new polymorph, we just need to compare our $\text{NaMnO}_{2-y-\delta}(\text{OH})_{2y}$ electrode with other Birnessite Na_xMnO_2 electrodes with similar nanosheet morphology. As shown in the copied Figure R1, Aurbach and Choi et al. prepared Na-Birnessite with similar

nanosheet morphology (Chem. Mater. 2015, 27, 3721-3725). Although their samples have similar ultra-thin nanosheet morphology, the cycle performance and rate performance of their samples are much worse as compared to those of our S4 sample. Even in the present work, S1 to S4 samples also have similar morphologies, especially for S3 and S4 samples, having exactly the same morphology. It is obvious that the S4 sample possesses greatly improved rate performance and cycling stability as compared to S1-S3 samples, demonstrating the outstanding electrochemical performance of the S4 sample is also attributed to its unique crystal structure with new polymorph. Therefore, in addition to the nanoscale morphology, the new polymorph of the S4 sample is effective to further improve its rate performance and cycle performance for Na storage. The outstanding rate performance and cycling stability of the $\text{NaMnO}_{2-y-\delta}(\text{OH})_{2y}$ electrode (S4) in our present study are the result of both nanostructure and new polymorph.

Redacted

Figure R1. Na-Birnessite with similar nanosheet morphology and its rate and cycle performance from literature (Chem. Mater. 2015, 27, 3721-3725).

The work by Aurbach and Choi et al. (Chem. Mater. 2015, 27, 3721-3725) has been added in Table 1 for comparison. A new paragraph with related discussion has been added at page 18 and 19 in the revised manuscript as follows:

“Obviously, the excellent rate performance and cycle performance of the present S4 sample should also be attributed to its ultra-thin nanosheet morphology, which effectively shortens the ion diffusion paths and mitigates structural variation.³³ As reported in literature¹⁰, Aurbach and Choi et al. prepared Na-birnessite with similar nanosheet morphology. Although their samples have similar ultra-thin nanosheet morphology, the cycle performance and rate performance of their samples are much worse as compared to those of the present S4 sample. Therefore, in addition to the ultra-thin nanosheet morphology, the new polymorph of the S4 sample is effective to further improve its rate performance and cycle performance for Na ion storage.”

2) Again, surface and subsurface redox reactions (pseudo-capacitive processes) may explain the small volume change occurred upon charging/discharging in this work.

Response: Thanks for the valuable comments. We agree with the reviewer that the surface and subsurface redox reactions could facilitate small volume change during charge/discharge processes. A new paragraph with related discussion has been added at page 21 in the revised manuscript as follows:

“Moreover, the surface and subsurface redox reactions of monoclinic $\text{NaMnO}_{2-y-\delta}(\text{OH})_{2y}$ sample with nanoscale morphology could also facilitate small volume change during charge/discharge processes.”

3) To confirm the existence of oxygen vacancies, in addition to the Mn L-edge spectrum, the O K-edge spectrum should be provided and the variations between the corresponding O pre edge and O K-edge main peak should be carefully compared and discussed.

Response: Thanks for the valuable comments. The O K-edge XANES spectra of S3 and S4 samples are provided in Figure R2. As shown in Figure R2a, the Mn L_{II,III}-edge shifts to lower energy for S4 as compared to that of S3, demonstrating decrease of average Mn valence in S4. Differences in the O K-edge fine structure can be observed between the S3 and S4 spectra as revealed in Figure R2b. The O K-edge pre peak is significantly reduced in the spectrum for S4 as compared to that of S3.

The decrease in intensity of the pre-peak can be ascribed to the reduction of neighboring Mn and formation of Oxygen vacancies (J. Am. Chem. Soc. 2017, 139, 4835-4845). Mn reduction comes along with oxygen vacancy formation because of charge compensation.

Figure R2. Mn L_{II,III}-edge (a) and O K-edge (b) XANES spectra of the S3 and S4 samples.

XANES results has been added as Supplementary Figure 3 in Supporting Information. A new paragraph with related discussion has been added at page 7 and 8 in the revised manuscript as follows:

“As shown in Supplementary Fig. 3a (Standard X-ray absorption near edge structure (XANES) spectra of Mn L_{II,III}-edges for MnO and Mn₂O₃ are also included as reference), the Mn L_{II,III}-edge shifts to lower energy for S4 as compared to that of S3, demonstrating decrease of average Mn valence in S4. Differences in the O K-edge fine structure can be observed between the S3 and S4 spectra as revealed in Supplementary Fig. 3b. The O K-edge pre-peak is significantly reduced in the spectrum for S4 as compared to that of S3. The decrease in intensity of the pre-peak can be ascribed to the reduction of neighboring Mn and formation of Oxygen vacancies¹⁷. Mn reduction comes along with oxygen vacancy formation because of charge compensation.”

Reviewer #2:

The authors positively answered to the questions raised. It is very nice to observe a retained morphology even after 1000 cycles at fast rate. I am still skeptical however about the quantification they provided using these XPS data. I also find it doubtful that the SEI can be observed by XRD, that would mean a very thick SEI, which is not apparent on the post mortem SEM images.

Despite these minor points, I recommend publication of the manuscript as all the points have been answered properly.

Response: Thank you very much for your good comments that help us greatly improve our manuscript.

REVIEWERS' COMMENTS:

Reviewer #1 (Remarks to the Author):

The reviewer has carefully read the authors's response and revised manuscript. Overall, Instead of providing more convincing evidences, the authors are dodging the raised questions on their claims that the ultrafast and ultrastable Na ion storage is induced by the newly designed crystal structure. Therefore, the reviewer believes that this paper is not acceptable at the present form.

Reviewer #1 (Remarks to the Author):

The reviewer has carefully read the authors's response and revised manuscript. Overall, Instead of providing more convincing evidences, the authors are dodging the raised questions on their claims that the ultrafast and ultrastable Na ion storage is induced by the newly designed crystal structure. Therefore, the reviewer believes that this paper is not acceptable at the present form.

Response: Thanks for the comments. According to the reviewer's comments and editor's instruction, we agree to soften the claim on the role of the new polymorph. The outstanding rate performance and cycling stability of the monoclinic $\text{NaMn}_{2-y}\delta(\text{OH})_{2y}$ electrode can be attributed to the combination of the nanoscale morphology and the new polymorph. Future work will be required to quantify and differentiate their contributions.